

# J-score: a robust measure of clustering accuracy

Navid Ahmadinejad, Yunro Chung and Li Liu

[1] Biodesign Institute, Arizona State University, Tempe, AZ, United States of America
[2] College of Health Solutions, Arizona State University, Phoenix, AZ, United States of America

## ABSTRACT

**Background**. Clustering analysis discovers hidden structures in a data set by partitioning them into disjoint clusters. Robust accuracy measures that evaluate the goodness of clustering results are critical for algorithm development and model diagnosis. Common problems of clustering accuracy measures include overlooking unmatched clusters, biases towards excessive clusters, unstable baselines, and difficulties of interpretation. In this study, we presented a novel accuracy measure, J-score, to address these issues.
**Methods**. Given a data set with known class labels, J-score quantifies how well the hypothetical clusters produced by clustering analysis recover the true classes. It starts with bidirectional set matching to identify the correspondence between true classes and hypothetical clusters based on Jaccard index. It then computes two weighted sums of Jaccard indices measuring the reconciliation from classes to clusters and *vice versa*. The final J-score is the harmonic mean of the two weighted sums.
**Results**. Through simulation studies and analyses of real data sets, we evaluated the performance of J-score and compared with existing measures. Our results show that J-score is effective in distinguishing partition structures that differ only by unmatched clusters, rewarding correct inference of class numbers, addressing biases towards excessive clusters, and having a relatively stable baseline. The simplicity of its calculation makes the interpretation straightforward. It is a valuable tool complementary to other accuracy measures. We released an R/jScore package implementing the algorithm.

## INTRODUCTION

Cluster analysis is an unsupervised data mining technique that partitions data into groups based on similarity (*Rodriguez et al., 2019*). It is a valuable approach to discover hidden structures and has broad applications in pattern recognition. Many clustering methods have been developed and data sets are subject to cluster analysis constantly (*Alashwal et al., 2019*; *Caruso et al., 2017*; *Saxena et al., 2017*). To evaluate algorithm performance, select models, and interpret partition structures, a robust measure of clustering accuracy is imperative.

Cluster analysis speculates that subsets of the input data belong to different classes and aims to discover these classes by partitioning data into hypothetical clusters. When true class labels of input data are known, accuracy of clustering results can be

Corresponding authors
Yunro Chung, yunro.chung@asu.edu
Li Liu, liliu@asu.edu

assessed on how well hypothetical cluster assignments recover true class labels (*Halkidi, Batistakis & Vazirgiannis, 2001*). Intuitively, the assessment involves first establishing the correspondence between true classes and hypothetical clusters (*i.e.,* set matching) (*Rezaei & Fränti, 2016*), then quantifying the overall goodness of match. For example, given a true class, the hypothetical cluster sharing the largest number of data points with it may be regarded as the best match. The fraction of total unmatched data points aggregated over all classes is then reported as an H-score (*Meilă & Heckerman, 2001*). The best cluster matched to a class can also be determined to maximize the harmonic mean of precision and recall rates (*i.e.,* F1-score). Weighting F1-scores over all classes produces an F-score to represent the overall accuracy (*Sundar, Chitradevi & Geetharamani, 2012*). However, because set matching reports only the best cluster matched to each class, "stray" clusters that are unmatched to any classes do not contribute to the final accuracy score. When two hypothetical partition structures differ solely by stray clusters, these accuracy measures are unable to distinguish them (*Meilă, 2007*; *Rosenberg & Hirschberg, 2007*).

To address this "problem of matching", several measures have been developed that circumvent set matching. Instead of creating specific class-cluster pairs, mutual agreement between all classes and all clusters is calculated based on data points that are consistently grouped together (mutual presence) or separately (mutual absence) in the two partition structures (*Cheetham & Hazel, 1969*). For example, Rand index RI (*Rand, 1971*) and its adjusted form ARI (*Hubert & Arabie, 1985*) search among all possible pairs of data points to find those that are mutually present or absent and calculate the ratios. Normalized mutual information (NMI) measures the geometric mean of entropy of the two partition structures (*Strehl & Ghosh, 2002*). Variation of information (VI) criterion and the normalized form NVI first compute the mutual information between the two partition structures, then derives the amount of information change when representing one partition structure with the other. V-measure uses conditional entropy to estimate skewness of data points in a cluster towards a class (homogeneity) or *vice versa* (completeness) and calculate the harmonic mean (*Rosenberg & Hirschberg, 2007*). However, information theoretic based measures have a known bias towards excessive small-sized clusters because a large number of mutually absent data points can inflate the amount of mutual information (*Amelio & Pizzuti, 2015*; *Lei et al., 2017*; *Vinh, Epps & Bailey, 2010*).

Identifying correspondence between classes and clusters is also helpful for algorithm evaluation and model diagnosis. For example, "stray" clusters unmatched to any classes and "split" classes matched to multiple clusters help pinpoint weaknesses in a clustering algorithm. Because set-matching-free measures do not provide such information, *post hoc* processing to identify class-cluster pairs needs to be performed, often using set-matching approaches these measures try to avoid at the first place. Given the disconnection, assessment based on set-matching-free measures may not align with interpretations of class-cluster pairs.

In this study, we introduce J-score, a novel clustering accuracy measure that supports four desirable properties. First, it performs set matching to identify correspondence between classes and clusters. Second, set matching is bidirectional, which finds clusters best aligned to classes and *vice versa*. Subsequent accuracy calculation thus incorporates

both matched and unmatched clusters to address the "problem of matching". Third, it is based on Jaccard index (*Cheetham & Hazel, 1969*) that is calculated using data points of mutual presence, avoiding the impact of mutual absence on accuracy assessment. Thus, it is robust to partition structures with excessive small-sized clusters. Fourth, its value is bounded between 0 and 1. We illustrate the behavior of J-score with theoretical proofs, extensive simulations, and applications to real benchmark datasets.

## MATERIALS & METHODS

Portions of this text were previously published as part of a preprint (*Ahmadinejad & Liu, 2021*).

### J-Score

Let $D$ represent a data set containing $N$ samples that are categorized into $T$ true classes. Clustering analysis of $D$ produces $K$ hypothetical clusters, each containing a disjoint subset of $N$. The class and cluster assignments of these samples are stored in a matrix $M$, which has $N$ rows and two columns. For a given sample in row $i$, the first column corresponds to the class label $t_i \in T$ and the second column corresponds to the cluster assignment $k_i \in K$. For a specific pair of class $t$ and cluster $k$, the sets $V_t$ and $V_k$ consist of samples for which $M_{i \in 1:N,1} = t$ and $M_{i \in 1:N,2} = k$, respectively. The Jaccard index is calculated as

$$I_{t,k} = \frac{|V_t \cap V_k|}{|V_t \cup V_k|} \tag{1}$$

where $|?|$ denote the size of a set.

### *Bidirectional set matching*

To establish the correspondence between $T$ and $K$, we first consider each class as the reference and identify its best matched cluster ($T \rightarrow K$). Specifically, for a class $t$ we search for a cluster $k$ that has the highest Jaccard index,

$$I_t = \max_{k \in K} \left( I_{t,k} \right). \tag{2}$$

We then reverse the direction of matching, *i.e.,* consider each cluster as reference and identify its best matched class ($K \rightarrow T$) using a similar procedure. For a cluster $k \in K$, we search for a class $t \in T$ with the highest Jaccard index,

$$I_k = \max_{t \in T} \left( I_{t,k} \right) \tag{3}$$

### *Calculating overall accuracy*

To quantify the overall accuracy, we aggregate Jaccard indices of individual clusters and classes, accounting for their relative sizes (*i.e.,* number of data points). We first compute a weighted sum of $I_t$ across all classes as $R = \sum_{t \in T} \left( \frac{|V_t|}{N} I_t \right)$, and a weighted sum of $I_k$ across all clusters as $P = \sum_{k \in K} \left( \frac{|V_k|}{N} I_k \right)$. We then take their harmonic mean as $J$ score,

$$J = \frac{2 \times R \times P}{R + P} \tag{4}$$

*Implementation*

We implemented $J$-score calculation in R language and released an jScore package in R/CRAN repository.

## Distribution of J-Score

**Proposition 1**. J-score is bounded between $(0, 1]$, with the maximum value of 1 reached when hypothetical clusters match true classes perfectly.

**Proof of Proposition 1**. Based on the theorem of Jaccard index (*Cheetham & Hazel, 1969*), $I_{t,k}$ in Eq. (1) ranges from 0 when $V_t$ and $V_k$ share no common data point (*i.e.,* $|V_t \cap V_k| = 0$) to 1 when data points in $V_t$ and $V_k$ overlap completely (*i.e.,* $|V_t \cap V_k| = |V_t \cup V_k|$). It is easy to derive that $I_t$ in Eq. (2) obtains the maximum value of 1 when class $t$ is paired with a perfectly matched cluster. Similarly, $I_k$ in Eq. (3) maximizes to 1 when cluster $k$ is paired with a perfectly matched class.

**Lemma 1.1** If all classes have perfectly matched clusters and *vice versa*, $R$ and $P$ in Eq. (4) equal to 1 regardless of the fraction of data points in each class and cluster, giving rise to $J = 1$.

**Lemma 1.2** If at least one class or cluster cannot find a perfectly matched counterpart, the corresponding $I_t$ and $I_k$ will be less than 1. Consequently, $R$ and $P$ will be less than 1, resulting in $J < 1$.

Because Lemma 1.1 and 1.2 exhaust all possible scenarios of bidirectional matches, J-score has a maximum value of 1 when classes and clusters have complete agreement. Meanwhile, bidirectional matching ensures that each and every class will be matched to a cluster sharing at least one data point, and vice versa. Therefore, $R$, $P$, and $J$ are all non-zero positive values.

**Proposition 2**. The baseline value of J-score is obtained from clustering results from a useless algorithm that randomly assigns data points into an arbitrary number of clusters. This baseline value is not constant.

**Proof of Proposition 2**. A useless clustering algorithm implies $M_{i,1}$ and $M_{i,2}$ are independent. By this definition, we derive the following lemmas.

Lemma 2.1. Denote $V_{t,k} = V_t \cap V_k$ as the set of data points that have class label $t \in T$ and belong to cluster $k \in K$. Because a useless clustering algorithm produces $M$ where $M_{i,1}$ and $M_{i,2}$ are independent, $\Pr(M_{i,1} = t, M_{i,2} = k) = \Pr(M_{i,1} = t) \cdot \Pr(M_{i,2} = k)$. As $\Pr(M_{i,1} = t) = \frac{|V_k|}{N}$ and $\Pr(M_{i,2} = k) = \frac{|V_t|}{N}$, the expected value of $|V_{t,k}| = N \cdot \frac{|V_k|}{N} \cdot \frac{|V_t|}{N} = \frac{|V_k||V_t|}{N}$.

Lemma 2.2. Denote $f(x) = \frac{ax}{bx+c}$ with $x, a, b, c > 0$. Because $\frac{d}{dx}f(x) = \frac{ac}{(bx+c)^2}$ is strictly greater than zero, $f(x)$ is a strictly increasing function.

For a given class-cluster pair $(t, k)$, the Jaccard index in Eq. (1) $I_{t,k} = \frac{|V_t \cap V_k|}{|V_t \cup V_k|} = \frac{|V_{t,k}|}{|V_t| + |V_k| - |V_{t,k}|}$. By Lemma 2.1, $I_{t,k}$ can be reduced to $\frac{|V_t||V_k|/N}{|V_t| + |V_k| - |V_t||V_k|/N} = \frac{|V_t||V_k|}{(N-|V_t|)|V_k| + N|V_t|}$. Lemma 2.2 shows $I_{t,k}$ strictly increases as $|V_k|$ increases. Thus, Eq. (2) $I_t = \max_{k \in K}(I_{t,k})$ is further reduced to $\frac{|V_t||V_{k\_max}|}{(N-|V_t|)|V_{k\_max}| + N|V_t|} \equiv I_{0t}$, where $|V_{k\_max}|$ is the maximum of $\{|V_{k \in K}|\}$.

Similarly, $I_k$ in Eq. (3) is reduced to $\frac{|V_k||V_{t\_max}|}{(N-|V_k|)|V_{t\_max}| + N|V_k|} \equiv I_{0k}$, where $|V_{t\_max}|$ is the maximum of $\{|V_{t \in T}|\}$. According to Eq. (4), $J_0 = \frac{2 \times R_0 \times P_0}{R_0 + P_0}$, where $R_0 = \sum_{t \in T}(\frac{|V_t|}{N}I_{0t})$, and

$P_0 = \sum_{k \in K}(\frac{|V_k|}{N}I_{0k})$. Because $R_0$ and $P_0$ are strictly positive, $J_0$ is strictly positive. $J_0$ is also a function of $\{|V_{k \in K}|\}$ and $\{|V_{tt \in T}|\}$ as $|V_{k\_max}|$, $|V_{t\_max}|$ are functions of $\{|V_{k \in K}|\}$ and $\{|V_{t \in T}|\}$.

$J_0$ is the J-score by comparing clustering results from a useless algorithm with the true class labels. $J_0$ varies by the number of true and hypothetical clusters and has no single value. For example, suppose that the $N$ data points are equally distributed across the total $|K|$ clusters and total $|T|$ true classes, *i.e.*, $|V_k| = N/|K|$ and $|V_t| = N/|T|$. Using the formula in Proposition 2, $J_0 = \frac{1}{|T|+|K|-1}$, which decreases as $|T|$ or $|K|$ increases. This means the minimum value of J-score varies even for the same data set. It maximizes when all data points are grouped into a single cluster. It minimizes when each data point is assigned to a separate cluster.

## Simulations to evaluate performance

To simulate an input data set $D$ with known class labels, we generated random numbers based on Gaussian distributions $G(\mu, 0.05)$ where $\mu$ is the mean and 0.05 is the fixed standard deviation. Data points generated from the same Gaussian distribution belonged to the same class. We then mixed data points of different classes to produce the input data. Class labels of these data points were ground truth.

Given an input data set $D$ with $N$ data points, we used three approaches to simulate a hypothetical partition structure. The first approach simulated a pre-determined partition structure, in which the total number of clusters $K$, the size of each cluster $N_k$, and the assignment of each data point to a cluster were specified manually. The second approach simulated a random partition structure. Here, only the value of $K$ was pre-specified. $N_k$ was determined by randomly choosing $K$ integers that summed to $N = \sum_1^K N_k$. Assignment of data points to clusters was also random, which was achieved by first repeating each $k$ value by $N_k$ times to create an ordered list of cluster labels, then permutating these labels. The third approach simulated splitting or merging classes. Given the pre-specified value of $K$, classes to be split or merged and the splitting ratio were randomly selected under the constraint that $N = \sum_1^K N_k$.

## Analysis of real datasets

We used two benchmark datasets to evaluate the performance of J-score and other accuracy measures. The first data set contains observations of 150 iris plants categorized into three true classes (50 Setosa, 50 Versicolor, and 50 Virginica) (*Anderson, 1935*). The second data set contains information of 178 wines categorized into three true classes (59 cultivar_1, 71 cultivar_2, and 48 cultivar_3) (*Aeberhard, Coomans & DeVel, 1994*). We performed K-means clustering analysis and agglomerative hierarchical clustering analysis of the scaled features. For K-means clustering, we used the kmeans() function in R with the default Hartigan-Wong algorithm, 10 random initialization sets (nstart =10), and 10 maximum iterations (iter.max =10), and tested a series of $K$ values from 2 to 10. For hierarchical clustering, we used the dist() function in R to calculate pairwise Euclidean distance, the hclust() function to build a dendrogram with the "ward.D2" algorithm, and the cutree() function with a series of $K$ values from 2 to 10 to create disjoint clusters. Given the true

classes in the input data and the clusters identified from clustering analysis, we computed various accuracy measures to find the best number of clusters that maximizes the accuracy. Ideally, the best number of clusters shall be 3 for both the Iris data set and the Wine data set. Deviation from the expected number indicates errors associated with a specific accuracy measure.

## Computation and comparison of various accuracy measures

We compared J-score with commonly used clustering accuracy measures. To compute ARI, RI, NMI, NVI, and V-measure, we used the R/aricode and R/clevr packages. To compute F- and H-score, we used an in-house developed R functions based on the published algorithms (*Meilă & Heckerman, 2001*; *Fung, Wang & Ester, 2003*).

## RESULTS

### J-Score addresses the "problem of matching"

The "problem of matching" arises when an accuracy measure fails to consider the presence of stray clusters in the evaluation process. J-score addresses this issue *via* bidirectional set matching, which enables the recovery of stray clusters. To illustrate the advantages of bidirectional over unidirectional set matching, we conducted simulation experiments in which the number of hypothetical clusters was set to be equal to, less than, or greater than the number of true classes. For each scenario, we measured the similarity between the true partition structure and the hypothetical partition structure using H-score and F-score, which utilizes unidirectional $T \rightarrow K$ matching, as well as J-score, which incorporates bidirectional matching.

For ground truth, we generated 100 random numbers $D_i$ ( $i = 1, \dots, 100$) belonging to three classes. Specifically, $D_{1,\dots,10}$ from a Gaussian distribution $G(1, 0.05)$ constituted class $T_a$, $D_{11,\dots,40} \in G(2, 0.05)$ constituted class $T_b$, and $D_{41,\dots,100} \in G(3, 0.05)$ constituted class $T_c$ (Fig. 1A). We first examined hypothetical partition structures that contained more clusters than classes. In one simulation, we grouped the data points into four clusters (Fig. 1B). Clusters $K_1$ and $K_2$ consisted of all data points from classes $T_a$ and $T_b$, respectively; Cluster $K_3$ consisted of $D_{41,\dots,80}$ that are two thirds of data points from $T_c$; and cluster $K_4$ consisted of the remaining data points from $T_c$. Because the number of hypothetical clusters exceeded the number of true classes, stray clusters were inevitable. Indeed, all three approaches despite using different $T \rightarrow K$ algorithms, identified $K_1$, $K_2$, and $K_3$ as the cluster best matched to $T_a$, $T_b$, and $T_c$, respectively, and left $K_4$ unmatched. The H-score (0.20) and the F-score (0.88) were calculated based on the goodness of $T_a \rightarrow K_1$, $T_b \rightarrow K_2$, and $T_c \rightarrow K_3$ matches, with no information contributed by $K_4$. The "problem of matching" occurred when we split the unmatched $K_4$ cluster into two clusters $K_{4.1}$ and $K_{4.2}$ (Fig. 1C). This splitting obviously reduced the overall accuracy of the partition structure. But H-score and F-score remained unchanged because $K_{4.1}$ and $K_{4.2}$ were stray clusters and did not contribute to the final scores. J-score rescued these stray clusters in the $K \rightarrow T$ matching step, identifying $T_c$ as the class best matched to $K_{4.1}$ and $K_{4.2}$. Then, all clusters and classes contributed information to the final J-score. As expected, the J-score dropped from 0.77 to 0.75 after the splitting.

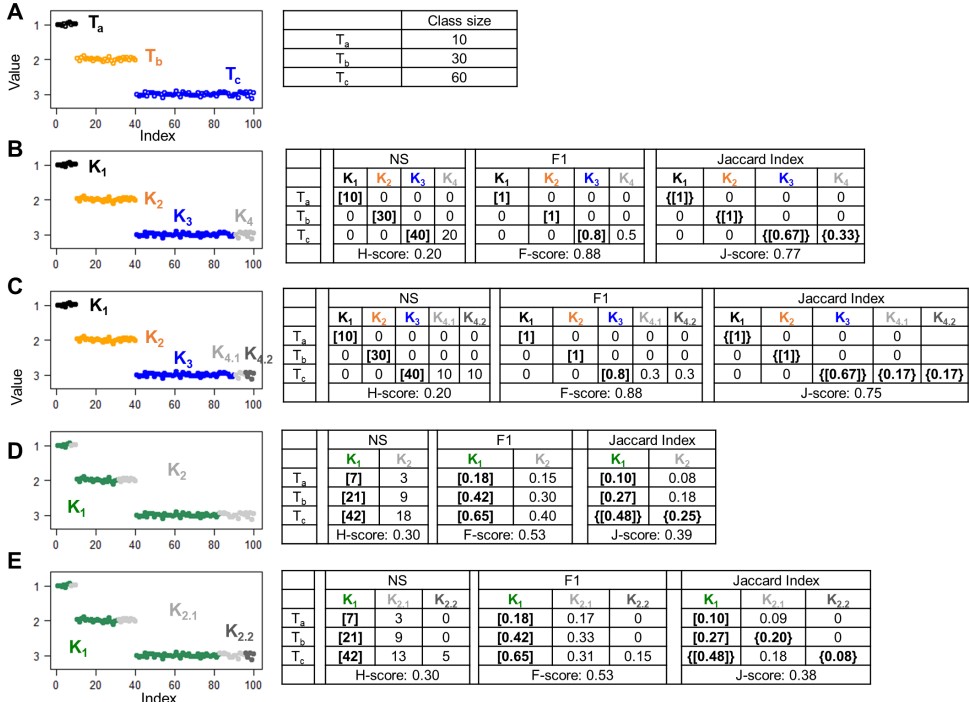

**Figure 1** **Simulations to illustrate the "problem of matching".** Scatter plots show 100 data points with indices ranging from 1 to 100 and values randomly generated from Gaussian distributions. Colors denote different classes or clusters. (A) The ground truth partition structure contains three classes. Class sizes are displayed in the table. (B–E) Hypothetical partition structures. Various scores and set matching results are displayed in the tables next the scatter plots. H-score and F-score perform unidirectional T to K matching based on the number of shared data points (NS) and F1 metric, respectively. J-score performs bidirectional T to K and K to T matching based on Jaccard Index. Square brackets indicate pairs inferred by T to K matching, where each class is paired with a cluster. Because this matching is anchored on classes, some clusters are not matched to any classes, thus "stray" clusters. Specifically, K4 in panel B, K4.1 and K4.2 in panel C, K2 in panel D and K2.1 and K2.2 in panel E are stray clusters. Curly brackets indicate pairs inferred by K to T matching where each cluster is an anchor and paired with a best match class.

Stray clusters exist even when there are fewer clusters than classes. In one simulation, we created a hypothetical partition structure consisting of two clusters. Specifically, cluster $K_1$ mixed 70% of the data points from each class. The remaining data points were grouped into cluster $K_2$ (Fig. 1D). After the $T \rightarrow K$ matching step, $K_1$ was repeatedly identified as the best matched cluster for all three classes, leaving $K_2$ as a stray cluster. In another hypothetical partition structure, we kept cluster $K_1$ untouched and split $K_2$ into $K_{2.1}$ and $K_{2.2}$ that were stray clusters as well (Fig. 1E). Again, because these two hypothetical partition structures differ only by stray clusters, H-score and F-score failed to distinguish them. J-score correctly reported a higher value for the first structure than the second structure (0.39 *vs.* 0.38).

## J-score reflects correct inference of class numbers

An important objective of cluster analysis is to infer the number of classes in input data. A robust accuracy measure shall reward a hypothetical partition structure if it contains the

correct number of clusters and penalize incorrect ones, as well as accounting for assignment of individual data points to each cluster. Using simulated data and real benchmark data sets, we examined how J-score and other measures varied with the number of clusters. Because H-score and NVI measure distances, we converted them to 1-H and 1-NVI, respectively, to measure similarities.

To minimize the confounding effect between class number inference and data point assignment, we simulated hypothetical partition structures by splitting or merging true classes. For ground truth, we generated 1,000 random numbers belonging to 10 classes. We varied the number of clusters $K$ from 1 to 50. For each hypothetical partition structure, we computed seven accuracy measures including J-score, F-score, V-measure, RI, ARI, NMI, and NVI. We repeated this process 200 times for each $K$ value and examined the mean and variance of each score. Since the number of true classes was 10, we expected these scores should peak at $K = 10$. This was indeed the case for J-score, F-score, ARI, and NMI (Fig. 2A). These scores also decreased sharply as $K$ deviated from 10, penalizing both deficient and excessive clusters. However, V-measure, RI, and NVI peaked incorrectly at $K = 13$, overestimating the number of classes (Fig. 2B). These two measures penalized excessive clusters only slightly. Even when $K = 50$ corresponding to overestimation by 400%, these scores decreased only by 5% from their peak values. To examine if J-score remains robust when the number of classes decreases and the class size increases, we simulated 1,000 random numbers belonging to five classes. Again, J-score peaked at correct inferences of class counts (Fig. 2C) while all the other scores overestimated (Fig. 2D). These results are consistent with previous reports of biases towards excessive clusters using existing accuracy measures (*Amelio & Pizzuti, 2015*; *Lei et al., 2017*; *Vinh, Epps & Bailey, 2010*). J-score is vigorous in this perspective.

We then examined the performance of these accuracy measures by applying them to clustering results of two real datasets. The first benchmark data set contains 150 iris plants belonging to three true classes. We performed K-means clustering of these plants. A key initial parameter of K-means is the number of clusters ($K$) to partition the data points into. Because $K$ is unknown in priori, we varied $K$ from 2 to 10. A robust accuracy measure is expected to maximize when $K$ is equal to the number of true classes. Indeed, J-score, along with ARI, RI, F-score, and NMI, peaked at $K = 3$, identifying the correct number of clusters. H-score, NVI, and V-measure maximized at $K = 2$, underestimating the number of clusters (Fig. 3A). To further examine the performance of these measures, we randomly sampled 90% of the Iris data set and repeated the above analysis 100 times. J-score, F-score, and NMI were the top performers, maximizing at $K = 3$ across all 100 tests (Fig. 3B). Conversely, H-score, NVI, and V-measure always underestimated the number of clusters, maximizing at $K = 2$ across all tests (Fig. 3C). ARI and RI made overestimations in 2% and 5% of the tests, respectively, maximizing at $K = 4$ or 5.

The second benchmark data set contains 178 wines belonging to three true classes. We performed agglomerative hierarchical clustering of these wines. Although hierarchical clustering does not need to know the value of $K$ when building dendrograms, this information is required when cutting the tree to create disjoint clusters. We again varied $K$ from 2 to 10. All measures except H-score maximized at $K = 3$ (Fig. 3D). We then

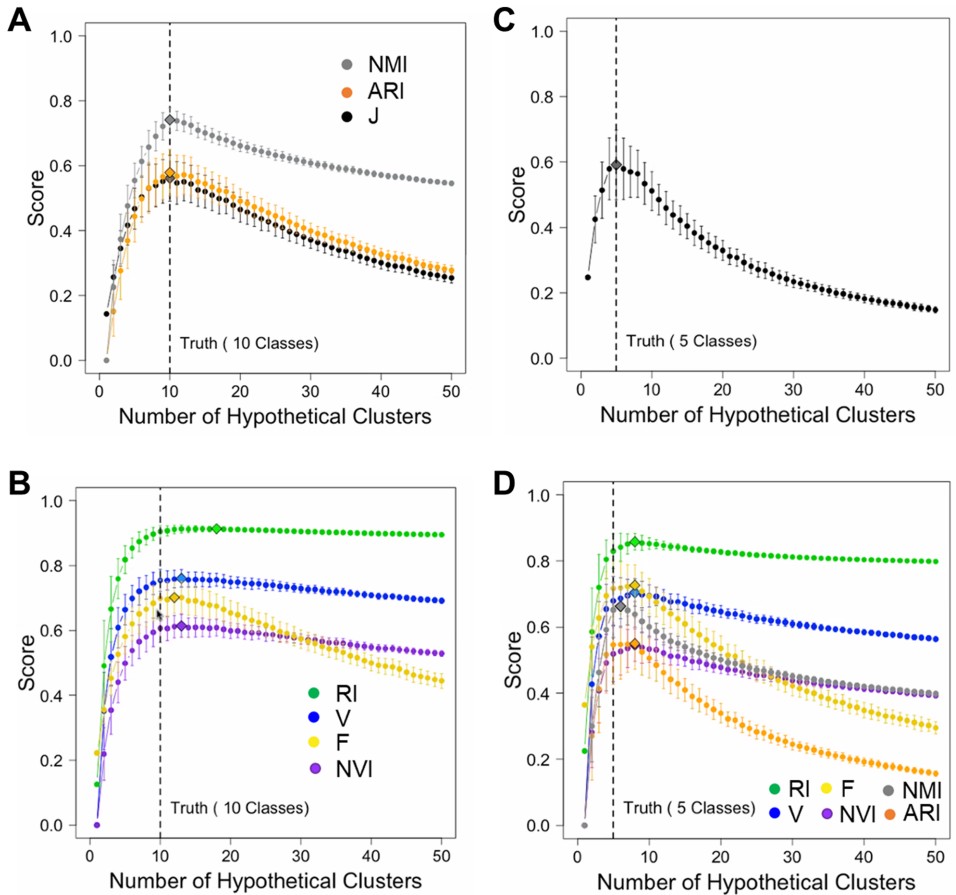

**Figure 2  Simulations to illustrate inferences of the number of classes.** (A, B) Simulations using a data set containing 1,000 data points from 10 true classes. For each accuracy measure, mean scores of 200 hypothetical partition structures containing a given number of clusters are displayed. Error bars represent standard deviations. Diamonds mark the inferred number of classes by the corresponding accuracy measures. J-score, NMR and ARI made correct inferences (A) and others made incorrect inferences (B). (C, D) Simulations using a data set containing 1,000 data points from five true classes. Only J-score made the correct inference (C) and others made incorrect inferences (D).

randomly sampled 90% of the Wine data set and repeated the analysis 100 times. J-score, NMI, NVI, and V-measure were the top performers, maximizing at $K = 3$ across all 100 tests (Fig. 3E). ARI, F-score, and RI made overestimations in 2–5% of the tests (Fig. 3F). H-score always made underestimations, maximizing at $K = 2$.

These results from analyzing simulated data and real data consistently supported that J-score is robust at inferring the correct number of clusters.

## J-score has a relatively stable baseline

The establishment of a baseline for an accuracy measure serves as a reference point to evaluate the performance of clustering algorithms and assess their ability to capture meaningful patterns in the data. It is important to note that the baselines of most clustering accuracy measures are not constant (*Vinh, Epps & Bailey, 2010*). In the proof of Proposition

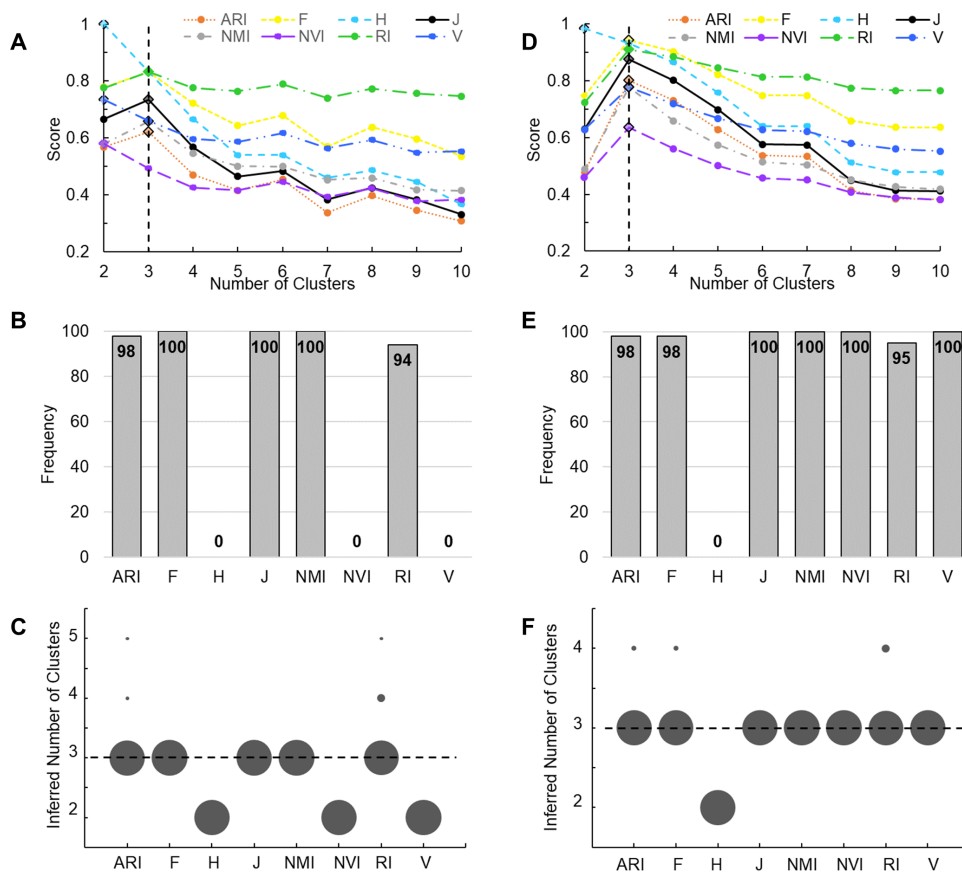

**Figure 3** **Evaluation using real datasets.** (A) K-means clustering analysis was applied to the complete Iris data set. Line plots show scores of various measures change with the number of clusters, *i.e.,* the K value. Solid lines represent measures that correctly peak at $K = 3$. Dotted lines represent measures that incorrectly peak at K ! = 3. The vertical line indicates $K = 3$. Diamonds mark the peak values. (B, C) A total of 100 K-means clustering analyses were applied, each to 90% of randomly sampled Iris data. Bar plot (B) shows the frequency of various measures correctly inferring three clusters. Dot plot (C) shows the frequency of the inferred number of clusters. Area of each dot is proportional to the frequency. (D) Agglomerative hierarchical clustering was applied to the complete Wine data set. Line plots show scores of various measures change with the $K$ value passed to the cutree function. The vertical line indicates $K = 3$. Diamonds mark the peak values. (E, F) A total of 100 hierarchical clustering analyses were applied, each to 90% of randomly sampled Wine data. Bar plot (E) shows the frequency of various measures correctly inferring three clusters. Dot plot (F) shows the frequency of the inferred number of clusters. Area of each dot is proportional to the frequency.

2, we showed that the baseline of J-score is also not constant. To further investigate this phenomenon, we examined how the baseline of each measure varies using simulated data sets and real data sets.

In the simulation experiments, we computed similarities between two random hypothetical partition structures. Specifically, we simulated 1,000 random numbers, randomly assigned them to $K$ clusters, and varied $K$ values from 2 to 50. For each $K$, we generated two random hypothetical partition structures and measured their similarities using various measures. We repeated this process 50 times for each $K$ value. Because these

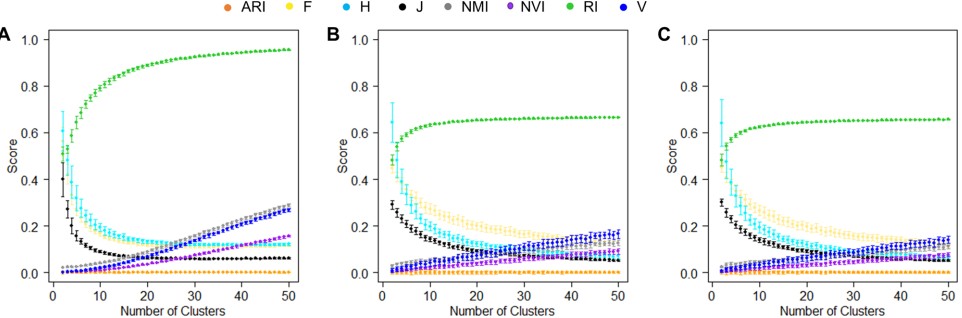

**Figure 4** **Baselines of various measures.** (A) Baselines estimated using simulation data. For each K, the mean score of 50 pairwise similarities between two random partition structures are displayed. Error bars represent standard deviations. (B, C) Baselines estimated using the real-world data sets, the Iris data (B) and the Wine data (C). For each K, the mean score of 50 pairwise similarities between a random partition structures and the true clusters are displayed.

partition structures were random, their mean pairwise similarities shall be close to the lowest value of the corresponding accuracy measure and stay constant regardless of the *K* value. This was indeed the case for ARI that is designed deliberately to maintain a stable baseline at 0 (Fig. 4A). The second best was J-score, with the baseline stabilized around 0.08 after the *K* value reached 12 and higher, *i.e.,* the mean cluster size was less than 83. However, when *K* was small, many data points were grouped together simply by chance, leading to high similarities between random partition structures and consequently high baseline values. Similar to J-score, the baselines of F-score and H-score quickly dropped as *K* increased but stabilized at a higher value (0.16). Conversely, the baselines of NMI, NVI, and V-measure steadily increased with the *K* value, showing no signs of stabilization across the entire range of K values tested. This pattern was consistent with the previous reports that these measures were biased towards overestimating the number of clusters (*Amelio & Pizzuti, 2015*; *Vinh, Epps & Bailey, 2010*). The baseline of RI looked the most different from the others, quickly approaching 1 as *K* increased.

We also examined the baselines of these accuracy measures in real datasets. Specifically, we randomly assigned the 150 iris plants (Fig. 4B) or 175 wines (Fig. 4C) to *K* clusters with *K* varied from 2 to 50. For each *K*, we generated a random clustering structure, measured its similarity to the true partition structure, and repeated this process 50 times. Unlike the simulated data where the two random clustering structures had the same value of *K*, this analysis using real data sets allowed the *K* value to be different from the number of true classes. The baselines of the various accuracy measures observed in this analysis were largely consistent with those observed in the simulated data. The only exception was RI, the baseline of which approached 0.6 instead of 1, reflecting the large impact of the numbers of clusters in two partition structures on this measure.

# DISCUSSION

In this study, we present J-score as a clustering accuracy measure that complements other existing measures. We illustrated the properties of J-score *via* theoretical inference, simulation studies, and analyses of real data sets.

The advantages of J-score over H-score and F-score measures are achieved *via* bidirectional matching to recover unmatched clusters. Although we did not compare other set matching algorithms such as those using distances between centroids to assign clusters to classes, the "problem of matching" applies because they are also based on unidirectional matching. To promote the adoption of bidirectional set matching, we included a utility function in our R/jScore package that takes in a table of arbitrary pairwise cluster/class similarity scores, performs bidirectional set matching, and returns the correspondences. This feature provides valuable information about stray clusters and split classes when diagnosing clustering results. In fact, because Jaccard index and F1-score are monotonically related, one can modify F-score using this utility function and potentially solve the "problem of matching" as well.

Compared to measures based on mutual agreement, J-score has three advantages. First, J-score identifies correspondence between classes and clusters and quantifies pairwise similarities, which is valuable information for model diagnosis and result interpretation. One can argue that *post hoc* set matching may be a remedy for mutual agreement-based measures. However, it does not fix the disconnect between matched sets and final accuracy scores, which can mislead model diagnosis and evaluation. Second, J-score does not suffer from biases toward excessive clusters. A partition structures with many small clusters risk overfitting. We show that J-score, without additional normalization or correction, performs equally well to NMI, better than or similar to ARI and NVI that involve complicated adjustment steps. Third, J-score has a relatively stable baseline when the average cluster size is not too big, making interpretation of a single accuracy value straightforward. Many existing measures lack this property except for ARI that specifically adjusts for chance (*Vinh, Epps & Bailey, 2010*). However, J-score achieves this without special adjustment, keeping the algorithm simple and intuitive.

Clustering analysis is an unsupervised learning method. Ground truth class labels are not available in real-world applications. In these cases, clustering accuracy can be estimated using internal validity measures such as the R2 score, the silhouette index, and the SDbw index that quantify within-cluster similarity and between-cluster similarity (*Liu et al., 2010*). Alternatively, one can generate two hypothetical partition structures of the same data set and assess their similarity using a clustering accuracy measure. Because high similarity implies high reproducibility, multiple hypothetical partition structures grouped on pairwise similarities can help identify the most reproducible ones. In fact, VI and NMI were originally developed to assess similarity between two hypothetical partition structures and later repurposed for comparing hypothetical and true partition structures. Furthermore, the additional information on matched class-cluster pairs, stray clusters, and split classes provided by J-score can be used to derive consensus or partial consensus clusters, and to identify the discrepant clusters for model diagnosis.

There are no perfect clustering accuracy measures. Like other available measures, J-score has its weaknesses. For example, it is not a true metric because it does not satisfy the triangular inequality. However, J-score has many desirable and complementary properties to existing measures, making it a valuable addition to the toolbox.

To ensure the reproducibility and scientific rigor of our results, we performed multiple repetitions of each simulation, ranging from 50 to 200 times, and reported the average performance along with the standard deviation. This approach allows for a robust assessment of the stability and consistency of the results obtained. For transparent reporting, we have made all source codes used in generating the simulation data and analyzing the simulated and real data sets available on the GitHub jScore site as well as at https://doi.org/10.5281/zenodo.8074044 (*Liu, 2023*). Other researchers and interested parties can access and verify the methods employed, reproduce the results, and build upon the work conducted in this study.

# CONCLUSIONS

J-score is a simple and robust measure of clustering accuracy. It addresses the problem of matching and reduces the risk of overfitting that challenge existing accuracy measures. It will facilitate the evaluation of clustering algorithms and clustering analysis results that are indispensable in big data analytics.

## Funding
This work was supported by the National Institutes of Health of USA (R01-LM013438). The funders had no role in study design, data collection and analysis, decision to publish, or preparation of the manuscript.

## Grant Disclosures
The following grant information was disclosed by the authors:
National Institutes of Health of USA: R01-LM013438.

## Competing Interests
The authors declare there are no competing interests.

## Author Contributions
- Navid Ahmadinejad conceived and designed the experiments, performed the experiments, analyzed the data, performed the computation work, prepared figures and/or tables, authored or reviewed drafts of the article, and approved the final draft.
- Yunro Chung conceived and designed the experiments, authored or reviewed drafts of the article, and approved the final draft.
- Li Liu conceived and designed the experiments, performed the experiments, analyzed the data, performed the computation work, prepared figures and/or tables, authored or reviewed drafts of the article, and approved the final draft.

## Data Availability

The source codes to compute J-score, generate simulation data, and analyze the simulated and real data sets is available at GitHub and Zenodo:

- https://github.com/liliulab/jscore
- Liu, Li. (2023). J-score: simulations and analyses presented in the manuscript. Zenodo. https://doi.org/10.5281/zenodo.8074044.

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
