# Peer review of "J-score: a robust measure of clustering accuracy"

_PeerJ Computer Science, doi:10.7717/peerj-cs.1545_

## Round 0.1 · original submission · Major Revisions

As you will see the reviewers have different opinions regarding Introduction section - try to improve it, to make it more mathematically solid. Further, you must put all efforts to present the proposed measures formally as required by the reviewer 3. You must test proposed measures on real datasets to demonstrate that it reflects the correct inference of the numbers of clusters and other useful properties described in the manuscript (take datasets from the publications mentioned by the reviewer 1 [1, 2]). The figures must be corrected in accordance with the requirements of the reviewers.

·

Basic reporting

The manuscript is clearly written in professional, unambiguous language and is well structured. It contains sufficient background. The code of the proposed algorithm to measure clustering accuracy (J score) is provided; however, synthetic datasets used for experiments are not provided, which may make it difficult for readers to reproduce the results of this work.

Experimental design

The research question is well-defined and relevant. I commend the authors for their extensive experiments on synthetic data. If there is a weakness, it is in the lack of numerical experiments on real datasets. It is common to test clustering algorithms and related measures not only on synthetic data but also on real ones (see, for example, [1, 2]). Please provide the results of J score applicability on real datasets to demonstrate that it reflects the correct inference of the numbers of clusters and other useful properties described in the manuscript.


1. R. Jenssen, K. E. Hild, D. Erdogmus, J. C. Principe and T. Eltoft, "Clustering using Renyi's entropy," Proceedings of the International Joint Conference on Neural Networks, 2003., Portland, OR, USA, 2003, pp. 523-528 vol.1, doi: 10.1109/IJCNN.2003.1223401
2. Aghagolzadeh, Mehdi, Hamid Soltanian-Zadeh, and Babak Nadjar Araabi. 2011. "Information Theoretic Hierarchical Clustering" Entropy 13, no. 2: 450-465. https://doi.org/10.3390/e13020450

Validity of the findings

no comment

Additional comments

1. There are a few typos (line 158: must be K4 instead of K2; line 175: must be F-score instead of J-score)
2. Unclear designation in parentheses in the table in Figure 1. If it indicates the best class-cluster pair, then it is not clear why Tc class best correlates with K4 cluster in Figure 1B. Please explain this notation in more detail.

Cite this review as

Reviewer 2 ·

Basic reporting

The point discussed in the line 70-72 about the set-matching free measures and the post hoc set matching needs to be more elaborated.
2. In line 79, it's written it mitigates excessive number of clusters (needs suitable explanation).

Experimental design

3. Experimentation can be carried out on some benchmark datasets also.
4. Results are shown only on synthetic datasets.
As clustering is unsupervised process, how the analysis can be carried out if the ground truth data is not available.

Validity of the findings

5. Figures can be more clearly drawn and explained.

Additional comments

The results need to be shown on a variety of datasets. Introduction need to be elaborated with addition to more literature.

Cite this review as
Anonymous Reviewer (2023) Peer Review #2 of "J-score: a robust measure of clustering accuracy (v0.1)". PeerJ Computer Science

Reviewer 3 ·

Basic reporting

The structure of definitions section is lacking as the important concepts and axioms are not highlighted. The notation is lacking especially in terms of indexing and set-subset structure.

The formal results such as corollaries, lemmas, and theorems are missing.

The discussion section is lacking scientific rigor.

Experimental design

Experimental setup is lacking in formal presentation and methodological justification.

Validity of the findings

Given the limited scope of experimental design and the lack of formal structure of the proposed clustering index, it is hard to asses if any conclusions could be reasonably drawn from the limited simulations.

Additional comments

1. Line 104: replace "truth clusters" with "ground truth clusters" or "true clusters" and explain how they relate to known classes.

2. There is a hanging parenthesis in line 88.

Cite this review as
Anonymous Reviewer (2023) Peer Review #3 of "J-score: a robust measure of clustering accuracy (v0.1)". PeerJ Computer Science

---

## Round 0.2 · accepted · Accept

I am happy to confirm that the authors responded positively to all comments and suggestions of the reviewers. The reviewers have not proposed any further changes, I decided to avoid any suggestions from myself as well. This new improved version may be accepted.

·

Basic reporting

no comment

Experimental design

I commend the authors for their additional experiments on real benchmark datasets.

Validity of the findings

no comment

Cite this review as